# TCU-Net: Transformer Embedded in Convolutional U-Shaped Network for Retinal Vessel Segmentation

**DOI:** 10.3390/s23104897

**Published:** 2023-05-19

**Authors:** Zidi Shi, Yu Li, Hua Zou, Xuedong Zhang

**Affiliations:** 1School of Electronic and Electrical Engineering, Wuhan Textile University, Wuhan 430077, China; 2School of Computer Science, Wuhan University, Wuhan 430072, China; 3School of Information Engineering, Tarim University, Alaer 843300, China

**Keywords:** retinal vessel segmentation, TCU-Net, efficient cross-scale transformer, channel cross-attention

## Abstract

Optical coherence tomography angiography (OCTA) provides a detailed visualization of the vascular system to aid in the detection and diagnosis of ophthalmic disease. However, accurately extracting microvascular details from OCTA images remains a challenging task due to the limitations of pure convolutional networks. We propose a novel end-to-end transformer-based network architecture called TCU-Net for OCTA retinal vessel segmentation tasks. To address the loss of vascular features of convolutional operations, an efficient cross-fusion transformer module is introduced to replace the original skip connection of U-Net. The transformer module interacts with the encoder’s multiscale vascular features to enrich vascular information and achieve linear computational complexity. Additionally, we design an efficient channel-wise cross attention module to fuse the multiscale features and fine-grained details from the decoding stages, resolving the semantic bias between them and enhancing effective vascular information. This model has been evaluated on the dedicated Retinal OCTA Segmentation (ROSE) dataset. The accuracy values of TCU-Net tested on the ROSE-1 dataset with SVC, DVC, and SVC+DVC are 0.9230, 0.9912, and 0.9042, respectively, and the corresponding AUC values are 0.9512, 0.9823, and 0.9170. For the ROSE-2 dataset, the accuracy and AUC are 0.9454 and 0.8623, respectively. The experiments demonstrate that TCU-Net outperforms state-of-the-art approaches regarding vessel segmentation performance and robustness.

## 1. Introduction

A large number of clinical studies have shown that diseases such as diabetic retinopathy (DR) [1], cataracts [2], dry eye syndrome (DES) [3], and glaucomatous lesions [4] are associated with structural and morphological alterations of retinal vessels. As part of ophthalmic diagnostic criteria, optical coherence tomography angiography (OCTA) enables the identification and measurement of blood flow to obtain high-resolution images of the blood vessels in the retina, choroid, and conjunctival areas [5]. Compared with traditional fluorescein fundus angiography and indocyanine green angiography, OCTA has the advantages of non-invasive, rapid, and three-dimensional imaging, making it a very promising vascular imaging technique in the field of ophthalmology [6]. As shown in Figure 1a, color fundus images obtained by conventional retinal imaging techniques have difficulty capturing fine vessels and capillaries. The optical coherence tomography angiography [7] techniques can generate images of the retinal vascular plexus at different depths in Figure 1b–d. High-quality OCTA images can present microvascular information in different OCTA depth layers, which can be easily applied to clinical research. To precisely identify and diagnose the variations in retinal blood vessels, medical personnel need to extract the retinal vessels from the fundus image to observe the length, curvature, width, and other morphological conditions of the retinal vascular trees. However, the manual segmentation of retinal vessels requires complicated work and is both tedious and time-consuming [8]. Various automatic segmentation algorithms that can improve efficiency and reliability have gradually attracted much attention in clinical practice procedures to solve this situation.

In the past few decades, many efforts have been made to segment retinal vessels. For instance, Gao et al. [9] proposed an automated method for the diagnosis of diabetic retinopathy that could help physicians diagnose patients more quickly and accurately. The approach relies on annotating a large number of images, which requires a lot of time and human resources, and reannotating images for different cases. Jin et al. [10] presented a new dataset of fundus images based on vascular segmentation, which can provide researchers with rich experimental data. The size of the dataset is not extremely large and includes only one disease (diabetic retinopathy), which may affect the generalization ability of the algorithm. Song et al. [11] presented a machine-learning-based clinical decision model that uses a set of rules developed by physician experts and combines traditional feature extraction methods with automatic feature learning by convolutional neural networks (CNNs) to improve the diagnostic accuracy of pathological ptosis. However, the study lacks comparative experiments to assess the advantages and disadvantages of the model with other methods. The state-of-the-art methods for retinal vessel segmentation come from the fully convolutional networks (FCNs), such as U-Net and its variants [12], which are based on the encoder–decoder architecture. U-Nets can capture contextual semantic information by using a cascade of convolutional layers and combining high-resolution feature maps with skip connections to achieve precise localization. The impact of skip connections is improved by Attention U-Net [13], which introduces an attention module to weight encoder features and fuse them with corresponding decoder features. This enhances the retention and reinforcement of critical vessel features in the decoder. However, the interactions between information at different scales are ignored by the skip connections, which only enhance the vessel representation by adding over the channels to the corresponding decoder features. It has been indicated by studies [14] that not all skip connections effectively connect the encoder and decoder. Additionally, it was found that the original U-Net performs worse than a U-Net without skip connections on some datasets.

Many studies have focused on retinal vessel segmentation in OCTA images due to the superiority of OCTA images in visualizing the retinal plexuses. OCTA images are characterized by rich retinal vessels, complex branching structures, and a low signal-to-noise ratio, making it difficult to distinguish small capillaries, arterioles, and venous regions in the image, which leads to poor segmentation. In addition, variety in vessel size, shadow artifacts, and retinal abnormalities further complicates segmentation. To address these challenges, Ma et al. [7] proposed a split-based coarse-to-fine OCTA image segmentation network (OCTA-Net) that comprises a coarse segmentation stage and a fine segmentation stage. The coarse segmentation network is utilized to generate preliminary confidence maps for pixel-level and centerline-level vessels, while the fine stage serves as a fusion network to obtain the final refined segmentation result. Although this approach divides OCTA image segmentation into two stages, mitigating the problem of discontinuity in vessel segmentation, the training process is laborious and impractical. Pissas et al. [15] presented an effective recurrent CNN for vessel segmentation in OCT-A, which uses fully convolutional networks (FCNs) to segment the entire image in each forward pass and iteratively refines the quality of vessel generation through weight-sharing coupled with perceptual losses. Despite achieving a good performance, CNN-based approaches generally exhibit limitations for capturing long-range (global) dependencies due to the intrinsic convolution operations. It causes the convolutional network to only focus on local features of the retinal vessel image, making it prone to breaking and missing the widely existing small blood vessels.

The existing studies have proposed that transformer architecture using the self-attention mechanism has emerged to make up for the information loss in convolution operations and effectively establish long-range dependencies. Self-attention is the key computational primitive of the transformer. It can implement pairwise entity interactions with a context aggregation mechanism, giving the transformer the ability to handle long-range dependencies. Preliminary studies with different forms of self-attention have shown its practicality in various medical image segmentation tasks [16,17]. Despite their exceptional representational power, the training and progress of the transformer architecture have intimidating challenges. One of the challenges is that complexity is quadratically related to the image input size in the vanilla transformer module. Secondly, without the ConvNet inductive biases, transformers cannot perform well on a small-scale dataset. The above challenges make it difficult to process a lesser number of medical images with higher resolutions, leaving a large amount of room for further improvements.

In summary, we have identified several limitations of existing OCTA retinal vessel segmentation methods: (1) The continuity of retinal vessels amplifies the defects of convolution calculations, and the convolutional network’s weak global capturing ability makes it susceptible to breaking or missing segmented vessels. (2) The skip connections in U-Net simply propagate vessel information from the encoder to the decoder on features of the same scale, resulting in limited interaction between features at different scales, which fails to prevent information loss and blurring. (3) Although the pure transformer network structures can achieve global context interaction through the self-attention mechanism, the high computational complexity of self-attention remains a challenge, especially for processing larger images with transformer-based structures.

To address these issues, this paper introduces a transformer embedded in a convolutional U-shaped network: TCU-Net, combining the advanced convolutional network and self-attention mechanism for OCTA retinal image segmentation. Specifically, an efficient cross-fusion transformer (ECT) is proposed to replace the original skip connections. The ECT module leverages the advantages of convolution and self-attention to avoid large-scale pre-training by exploiting the image induction bias of convolution, as well as the capability of the transformer to capture long-range relationships with linear computational complexity. Moreover, features with different scales are input by the encoder into an efficient multihead cross-attention mechanism to achieve interaction between different scales and compensate for the loss of vessel information. Finally, the efficient channel-wise cross attention (ECCA) module is introduced to fuse the transformer module’s multiscale features and decoder features to solve the semantic inconsistency between them and enhance effective vessel features. The main contributions of this work include the following:We proposed a novel end-to-end OCTA retinal vessel segmentation method that embeds convolution calculations into a transformer for global feature extraction.An efficient cross-fusion transformer module was designed to replace the original skip connections, thus achieving interaction between multiscale features and compensating for the loss of vessel information. The multihead cross-attention mechanism of the ECT module reduces the computational complexity compared to the original multihead self-attention mechanism.To reduce the semantic difference between the output of ECT module and decoder features, we introduce a channel cross-attention module to fuse and enhance effective vessel information.Experimental evaluation on two OCTA retinal vessel segmentation datasets, ROSE-1 and ROSE-2, demonstrates the effectiveness of the proposed TCU-Net.

## 2. Related Studies

The retinal vessel segmentation studied and considered herein can be divided into a CNN-based method and a transformer-based method. Among them, the transformer-based method focuses on its application to medical image datasets. In this section, we introduce corresponding algorithms for each category.

### 2.1. Based on Convolution Neural Networks

In recent years, deep learning models have been greatly used for retinal images since they do not need any handcrafted features and outperform existing unsupervised methods. Such models, especially U-Net [18], are still the most popular segmentation frameworks applied to fundus images up to now. Due to the blurred state of small blood vessels located at the end of blood vessels and the edges of blood vessels in retinal vascular images, as well as the unclear distinction between the blood vessel area and the background area, it is difficult to achieve accurate segmentation. To solve this issue, Xiao et al. [19] introduced the residual structure and combined it with U-Net to achieve a powerful feature extraction capability to obtain high-accuracy retinal vessel segmentation. However, ResU-Net [20] utilizes more convolutional layers and parameters, which might be overfitted. Guo et al. [21] introduced the spatial attention module to make the network focus on the vascular features and inhibit the unnecessary features, thus improving the expression ability of the network. As the attention module of SA-Unet only focuses on local information, that means it is not sensitive enough to long-range dependencies. Zhang et al. [22] proposed pyramid U-Net, which was employed in both the encoder and decoder to aggregate features at higher and lower levels for accurate retinal vessel segmentation. In this way, contextual information sharing and aggregation from coarse to fine can be achieved, thus improving the segmentation of capillary regions.

With the widespread use of OCTA techniques in ophthalmic diseases, researchers have gradually switched their targets from color fundus images to OCTA retinal vessel segmentation. Li et al. [23] proposed a new image magnification network (IMN) with a structure of an upsampling encoder and then a downsampling decoder. This design is to capture more image details and reduces the omission of thin-and-small structures. Xu et al. [24] introduced an OCTA-based cascaded neural network to automatically segment and distinguish small blood vessels before and after the capillary plexus, followed by a graph neural network (GNN) to improve the connectivity of the initial segmentation. Wu et al. [25] proposed a progressive attention-enhanced network (PAENet) for 3D-to-2D retinal vessel segmentation. It consists of a 3D feature learning path and a 2D segmentation path. To obtain more detailed information, a feature fusion module (FFM) is designed to inject 3D information into the 2D feature path and then model the semantic relationship between spatial and channel dimensions to achieve feature interaction. The above CNN-based segmentation networks achieved great performance in the retinal vasculature, but the local and limited receptive field of the convolutional network is still one of its shortcomings. Moreover, the existing U-Net-based retinal vascular segmentation networks only fuse features for the same level of encoder and decoder, ignoring the correlations between features of different layers. Therefore, the method proposed here interacts with the encoder features of different scales of U-Net to compensate for the loss of vascular information.

### 2.2. Based on Transformer Architecture

One of the first transformer-based architectures proposed for medical image segmentation is the TransUnet [26] architecture, which regards a hybrid CNN-transformer architecture as an encoder and outputs the final segmentation mask in the decoder. Zhang et al. introduced TransFuse [27] to effectively integrate the transformer and CNN features through the BiFusion module utilizing self-attention and a multimodal fusion mechanism. It was evaluated for polyp segmentation, skin segmentation, and hip segmentation and has been shown to be effective. In other work, TransAttUNet [28] is the first network to apply transformer layers between the encoders and decoders in a U-shaped architecture. The robust self-aware attention module and multiscale skip connection have been embedded between the encoder and decoder of U-Net, which not only enhances the flexibility of U-Net but also increases the expression ability of global spatial attention and transformer self-attention. Plenty of experiments with TransAttUNet on five benchmark medical image segmentation datasets have shown its effectiveness. The above transformer-based model implements global context modeling and exhibits a strong ability to capture key features in images. Nevertheless, the computational complexity of the original self-attention is high and requires a longer training time and a larger amount of computational resources. To address this, Tan et al. [29] proposed a novel transformer network (OCT2 Former) for OCTA retinal vessel segmentation, using a dynamic token aggregation transformer to reduce the huge computational overhead of the original transformer and designing an assisted convolution branch to speed up the convergence of the transformer. In addition, Guo et al. proposed a UTNet [30] model in which transformer layers are present in both the encoder and decoder. It effectively combines the attention mechanism with convolution operations and reduces the quadratic complexity of the self-attention mechanism to a linear type, respectively. In order to accelerate the convergence of the segmentation network, we reduce the computational complexity of the model by using the latter scheme to embed the features into the self-attentive mechanism after reducing their size through convolutional computation.

## 3. Proposed Method

### 3.1. Network Architecture

Figure 2 provides an overview of the TCU-Net network. The U-Net architecture comprises a downsampling encoder and an upsampling decoder, and the skip connections refer to adding encoder and decoder features at symmetric positions on the channel, thus preserving the original input feature map in the deep transformation. Inspired by methods such as UTNet [30] and UCTransNet [14], we aim to improve the performance of U-Net by designing an efficient cross-fusion transformer to replace the original skip connections. The ECT module is situated on the original skip connection structure. The output of the ECT module is not directly added to the channel with the corresponding layers of the decoder. Instead, it is fused with the output features and upsampled features layer-wise by the ECCA module. This process guides the decoder stage and enhances vascular information.

### 3.2. ECT: Efficient Cross-Fusion Transformer for Encoder Feature Transformation

To solve the high computational complexity issue when fusing the multiscale features of encoders, the proposed efficient cross-fusion transformer (ECT) module integrates convolution into the self-attention mechanism to avoid the large-scale pre-training of the transformer. This is attributed to the theory proposed by wang et al. [31] that self-attention is essentially low rank for long sequences and most of the information is concentrated on the largest singular value. A more efficient attention mechanism based on this theory was proposed by UTNet [30], which successfully reduced the computational complexity of self-attention. In addition, UCTransNet [14] identified that some skip connections may not be effective due to the incompatible feature sets between the encoder and decoder stages. To address this issue, they introduced the CTrans (channel transformer) model as an alternative to U-Net skip connections. The CTrans model effectively solves the semantic gap and achieves the accurate automatic segmentation of medical images. Inspired by them, the ECT module can effectively fuse features at different scales as well as reduce the computational complexity of the self-attention mechanism in Figure 3.

In previous studies [32], we calculated the attention function for a set of queries simultaneously, packed into a matrix **Q**. The keys and values are also packed into matrices **K** and **V**. We use 4 heads and consider an input feature map $X\in R^{C\times H\times W}$, where *H*, *W* is the spatial height and width, and *C* is the number of channels. The computation process is described as follows: (1)$$Attention\left(Q,K,V\right)=softmax\left(\frac{QK^{T}}{\sqrt{d}}\right)V,$$
where the $Q,K,V\in R^{d\times H\times W}$ and *d* is the embedding dimension of each head. Accordingly, the Q,K, and V are flattened and transposed into sequences with size $R^{n\times d}$, and n=HW. Consequently, the dot-product attention leads the complexity to $O(n^{2}d)$. Typically, self-attention layers are slower than recurrent layers when the sequence length *n* is longer than the representation dimensionality *d*, affecting the self-attention’s flexible applicability. Therefore, the main idea of the effective cross-fusion self-attention we employed is embedding the projection into the lower dimension.

In the efficient cross-fusion transformer (ECT) module, for each output $X_{i}\in R^{\frac{HW}{i^{2}}\times C_{i}}$, i=(1,2,3,4) of the encoder, it needs to be regularized to $X_{i}^{\prime }\in R^{\frac{HW}{i^{2}}\times C_{i}},i=\left(1,2,3,4\right)$ before entering the attention mechanism. As shown in Figure 3, we use three 1×1 convolutions to project $X_{i}$ into $Q_{i},K_{i},V_{i}\in R^{\frac{HW}{i^{2}}\times C_{i}},i=\left(1,2,3,4\right)$ and concatenate the four layers of K,V as the ultimate key and value $K_{\Sigma }=Concat\left(K_{1},K_{2},K_{3},K_{4}\right)$, $V_{\Sigma }=Concat\left(V_{1},V_{2},V_{3},V_{4}\right)$. On each of these projected versions of queries, keys, and values we then perform three projections to project them into low-dimensional embedding in each head: $Q^{\prime }\in R^{k\times d_{i}},K^{\prime }\in R^{k\times d_{\Sigma }}$, and $V^{\prime }\in R^{k\times d_{\Sigma }}$, i=(1,2,3,4), where *d* is the dimension of embedding in each head, $k=hw\leq \frac{HW}{i^{2}}$, and *h* and *w* are the reduced size of each feature map after a bilinear interpolation.

The proposed module contains six inputs containing four queries and two aggregated $K_{\Sigma }$, $V_{\Sigma }$ as the key and value, as shown in Figure 4. We compute the matrix of outputs through an efficient cross-attention (ECA) mechanism as: (2)$$ECA_{i}=\mathrm{softmax}\left(\frac{Q_{i}^{\prime T}K^{\prime }}{\sqrt{d_{\Sigma }}}\right)V^{\prime T},$$
where $d_{\Sigma }=\mathrm{Concat}\left(d_{1},d_{2},d_{3},d_{4}\right)$ is aggregated through the dimensions in the four skip connection layers. Finally, we computed the dot products of the transpose of the query with all keys, divide each by $\sqrt{d_{\Sigma }}$, and apply a softmax function to obtain the weights on the values. In practice, we use 4 heads and employ $k=\frac{HW}{16^{2}}$ as the limited length. Due to the reduced size of each feature map, the total computational complexity is similar to $O\left(k^{2}d\right)$ and much smaller than $O\left(n^{2}d\right)$.

To distinguish our model from conventional vision transformer models, we perform a convolutional layer for each output of the multiheaded self-attention, accompanied by a batch normalization and a ReLu activation function to achieve information complementarity. Hereinafter, applying a convolution calculation and residual structure, the output is obtained as follows: (3)$$E_{i}=X_{i}+ECA_{i}+\mathrm{Conv}\left(\mathrm{Relu}\left(\mathrm{Bn}\left(X_{i}+ECA_{i}\right)\right)\right),$$

The operation in Equation (Equation 3) is repeated four times to build the outputs of the transformer. Finally, we use an upsampling followed by a 1×1 convolution to reconstruct the four outputs $E_{1}$, $E_{2}$, $E_{3}$, and $E_{4}$ and splice them with the decoder features $D_{1}$, $D_{2}$, $D_{3}$, and $D_{4}$, respectively.

#### ECCA: Efficient Channel Cross-Attention

To solve the semantic inconsistency between the effective transformer and U-Net decoder, we apply a channel cross-attention module [14] by exploiting the inter-channel relationship of features. To compute the channel cross-attention efficiently, we firstly squeeze the spatial dimension of the input features $E_{i}\in R^{C_{i}\times H\times W}\;\mathrm{and}\;D_{i}\in R^{C_{i}\times H\times W}\left(i=1,2,3,4\right)$, respectively. For aggregating spatial information, average-pooling and max-pooling have been commonly adopted so far. In previous studies, we argued that max-pooling can gather the unique object features to infer finer channel-wise attention and average-pooling can learn the extent of the target object effectively [33]. Thus, we empirically confirmed that exploiting both of them in a parallel or sequential manner obtains the best result (see Section 4.1). We describe the computational process as follows: (4)$$M_{i}\left(E_{i}\right)=\sigma \left(\mathrm{Avg}\mathrm{Pool}\left(E_{i}\right)\right)$$
(5)$$M_{i}\left(D_{i}\right)=\sigma \left(\mathrm{Avg}\mathrm{Pool}\left(D_{i}\right)\right)$$
(6)$$N_{i}=L_{1}\cdot M_{i}\left(E_{i}\right)+L_{2}\cdot M_{i}\left(D_{i}\right)$$
(7)$$E_{i}^{\prime }=\sigma \left(N_{i}\right)\cdot E_{i}$$
where σ denotes the sigmoid function, $M_{i}\left(E_{i}\right)\in R^{C_{i}\times 1\times 1}\;,\;\mathrm{and}\;M_{i}\left(D_{i}\right)\in R^{C_{i}\times 1\times 1}$. Note that $L_{1}\in R^{C_{i}\times C_{i}}$ and $L_{2}\in R^{C_{i}\times C_{i}}$ are the weights of two linear layers. Through these computations, we generate two different pieces of spatial context information and merge the features using element-wise summation. Finally, the channel attention map is built by a single linear layer and sigmoid function.

## 4. Experimental Results

### 4.1. Datasets and Metrics

To evaluate the effectiveness and superiority of TCU-Net, we have conducted extensive experiments on the Retinal OCTA SEgmentation (ROSE) dataset [7], which is the first public ROSE dataset for the vessel segmentation task. ROSE consists of two subsets (ROSE-1 and ROSE-2) obtained by two different devices. To be specific, there are 117 OCTA images with a resolution of 304 × 304 pixels in ROSE-1, while ROSE-2 contains 112 OCTA images with 512 × 512 pixels. ROSE-1 can be divided into three kinds of OCTA images with both centerline-level annotation and pixel-level annotation, i.e., SVC, DVC, and SVC+DVC. In ROSE-2, only SVC images with centerline-level annotation are provided. We considered the consensus of centerline-level annotation and pixel-level annotation as the ground truth in the SVC of ROSE-1. Given a predicted segmentation result and its corresponding ground truth, true positives (TPs) mean the correctly segmented vessel pixels and those wrongly classified as non-vessel pixels are denoted as false negatives (FNs). Similarly, true negatives (TNs) mean correctly segmented non-vessel pixels and those incorrectly detected as vessel pixels are denoted as false positives (FPs). The evaluation metrics are calculated as follows:Area under the ROC curve (AUC) [34];Sensitivity (SEN) = TP/(TP + FN);Specificity (specificity) = TN/(TN + FP);Accuracy (ACC) [35] = (TP + TN)/(TP + TN + FP + FN);Kappa score [36] = (accuracy − pe)/(1 − pe);pe [7] = ((TP + FN)(TP + FP) + (TN + FP)(TN + FN))/(TP + TN + FP + FN)${}^{2}$False discovery rate (FDR) [37] = FP/(FP + TP);G-mean score [38] = sensitivity × specificity;Dice coefficient (Dice) [39] = 2 × TP/(FP + FN + 2 × TP).

### 4.2. Implements Details

We implemented the proposed method with PyTorch on an NVIDIA TITAN GPU and empirically set the number of epochs to 50 epochs for ROSE-1 and 300 epochs for ROSE-2. The stochastic search strategy was used to find the optimal hyperparameters, and after constant iterations of training, the best combination for the model was identified. We finally used Adam optimization to adaptively adjust them with a learning rate of 0.0006, a batch size of two, and a weight decay of 0.0001. Each kind in ROSE-1 is composed of 30 training images and 9 testing images, while 90 images in ROSE-2 are used for training, and the remaining 22 images are chosen for testing. Only when training, the random rotation of an angle of −10 and 10 is conducted for data augmentation. The poly learning rate policy with a poly power of 0.9 is adopted for better performance and stable training. It is worth noting that we train TCU-Net in an end-to-end manner with binary cross-entropy loss. To simplify the training process, we utilized the ground truth instead of centerline-level annotation and pixel-level annotation for ROSE-1.

### 4.3. Performance Comparison and Analysis

To comprehensively prove the superiority of the proposed method, we have compared it with many other state-of-the-art segmentation methods: seven CNN-based deep learning approaches—U-Net [18], ResU-Net [20], CE-Net [40], CS-Net [41], and OCTA-Net [7]—and two transformer-based deep learning networks—TransFuse [27] and TransUnet [26]. We report the objective metrics of these methods in Table 1, Table 2, Table 3 and Table 4 and subjective results in Figure 5. The network’s vascular segmentation ability can be observed from the ground truth comparison with the predicted mask.

**Subjective comparisons**. Figure 5 compares the resulting images of three advanced vascular segmentation methods, including two networks based on transformers for medical image segmentation. It can be observed that the two transformer networks have several vascular breakpoints in their prediction plots. Meanwhile, the OCTA-Net [7] outperforms the other two networks except for our proposed method, but it achieves weak performance in capturing thin vessels due to convolutional limitations. In contrast, the proposed method (TCU-Net) identifies more complete vessels without separate training of coarse and fine vessels and performs a more sensitive and accurate segmentation of capillaries. The graph of SVC and DVC vessel results in ROSE-1 demonstrates that TCU-Net is quite coherent in terms of overall vessels with minimal truncation points, and the results are better than the other three networks’ segmentation results, especially on the fine capillaries. Similar results are demonstrated in the ROSE-2 dataset. In the following, we will analyze the proposed method’s objective metrics.

Results of the SVC dataset in ROSE-1. We first evaluate the performance of the proposed method on the SVC dataset against a variety of SOTA methods in terms of the above evaluation metrics. As the experimental results in Table 1 show, TCU-Net outperforms CNN-based SOTA methods by a large margin and all metrics evaluated show the best performance. Specifically, compared to other transformer-based methods, TCU-Net also shows a superior learning ability on the majority of vessels. The performance of the proposed method is consistent with the segmentation results, demonstrating a strong connectivity and integrity in both coarse and fine vessels.

**Results of the DVC dataset in ROSE-1**. For the DVC images, their ground truth contains only the intermediate fine vessels. Our method shows the same optimal performance in fine vessel segmentation in Table 2. It is commendable that all objective metrics are higher than the latest methods; in particular, the mean value of the AUC is up to 98.23%, with an improvement of 1.41%, respectively, and a reduction of about 17.73% in FDR as compared to OCTA-Net. This result shows that TCU-Net is more sensitive to capillaries compared to other methods.

**Results of the SVC+DVC dataset in ROSE-1**. Each image of this dataset contains both SVC and DVC vascular maps. We repeat the experiments for U-Net and its variants several times again. The results are shown in Table 3, these prove that TCU-Net achieves state-of-the-art performance. Specifically, compared to CS-Net, the proposed network improved 0.21%, 0.3%, and 0.68% in the three metrics of the AUC, ACC, and Kappa, respectively, and reduced 2.22% in FDR. Table 3 and Table 5 show that the proposed method not only outperforms the two transformer frameworks but also can effectively reduce the computational complexity of the original transformer model and the number of parameters of the model.

**Results of the ROSE-2 dataset**. The difference between ROSE-2 and ROSE-1 is that ROSE-2 has a high pixel size of 512 × 512. Due to the high pixel count of the images, training on this dataset converges more slowly compared to ROSE-1. Therefore this dataset needs 300 epochs of training on the TCU-Net network to obtain the best value. As shown in Table 4, the proposed method achieves the best results on the AUC, ACC, G-mean, and Kappa, respectively. This result demonstrates that the TCU-Net network is equally adapted to high-pixel fundus image segmentation with the introduction of a self-attention mechanism.

### 4.4. Ablation Studies

In this paper, we conduct an ablation study to assess the effectiveness of the proposed method. Experiments are conducted to evaluate the effectiveness of the proposed branched design by choices of different attention combination schemes. The ROSE dataset that has been used and the results of the experiments are recorded.

**Ablation for the proposed modules**. To perform a thorough evaluation of the ECT module and the ECCA module, we added each component to U-Net, and the performance results are shown the by applying each component to the original scheme in Table 6, Table 7, Table 8, and Table 9 for the SVC, DVC, SVC+DVC, and ROSE-2 datasets, respectively. The performance of all datasets is improved by both the ECT module and the ECCA module. Specifically, the efficient cross-fusion transformer module successfully fuses multiscale features, leading to significant performance improvements and preventing information loss from the encoder. Furthermore, the ECCA module enhances performance by establishing an effective connection to the decoder features, thereby reducing ambiguity. Note that both types of attention are crucial, and the ‘Base+ECT+ECCA’ approach achieves the best values on all metrics, driving the performance of retinal vessel segmentation.

**Ablation for the projection of efficient self-attention and to reduce size**. Figure 6 and Figure 7 show the comparison of the Dice scores when the dimensions H and W of the feature map are reduced to 1/16, 1/8, and 1/4 of the original size. Among them, using interpolation downsampling is slightly better than using maximum pooling, and the best results are obtained by reducing the size of the ROSE-1 and ROSE-2 datasets to 1/4 and 1/16, respectively. In addition, we compare in terms of the model size and floating point of operations. As shown in Table 5, the proposed model has a substantial reduction in the number of parameters compared to the other transformer model, along with a significant performance improvement. This indicates that the proposed model shows superiority in vessel segmentation.

## 5. Conclusions

In this paper, we present a novel strategy to combine a transformer and U-Net for retinal vessel segmentation. Transformers are knowns as architectures with strong innate self-attention mechanisms. To enhance the effective vascular information, we propose an ECCA module to fuse the ECT module features with the decoder features. The proposed approach has a lower memory occupation and computational complexity than other transformer-based models [26,27], without pre-training. Nevertheless, it is crucial to emphasize that the clinical application of TCU-Net should be carefully evaluated by medical professionals due to potential variations in real images, such as illumination, shooting angles, and lesion areas. The proposed TCU-Net architecture achieves a state-of-the-art performance for ROSE-1 and ROSE-2 on SVC and DVC datasets, but further research is needed to address potential biases in practice. Future research could further explore and improve this approach to address potential biases in clinical practice and facilitate the model’s widespread use in clinical applications.

## Figures and Tables

**Figure 1 sensors-23-04897-f001:**
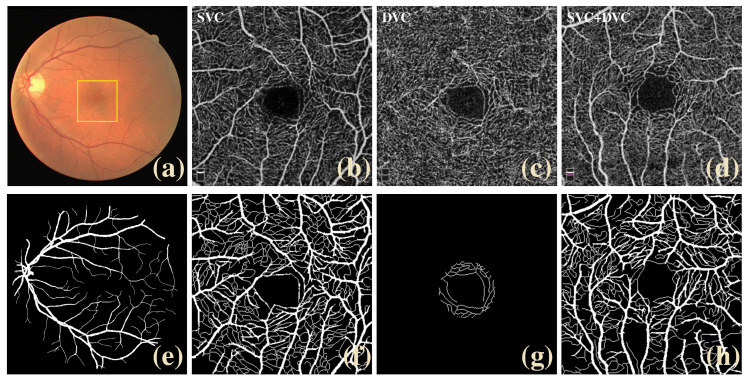
Comparison of color fundus images and fovea-centred (yellow rectangle area) OCTA images: (**a**) color fundus, (**b**–**d**) superficial vascular complexes (SVC), deep vascular complexes (DVC), and the inner retina vascular plexus including both SVC and DVC (SVC+DVC). (**e**–**h**) are their corresponding labels. The small vessels usually have low contrast.

**Figure 2 sensors-23-04897-f002:**
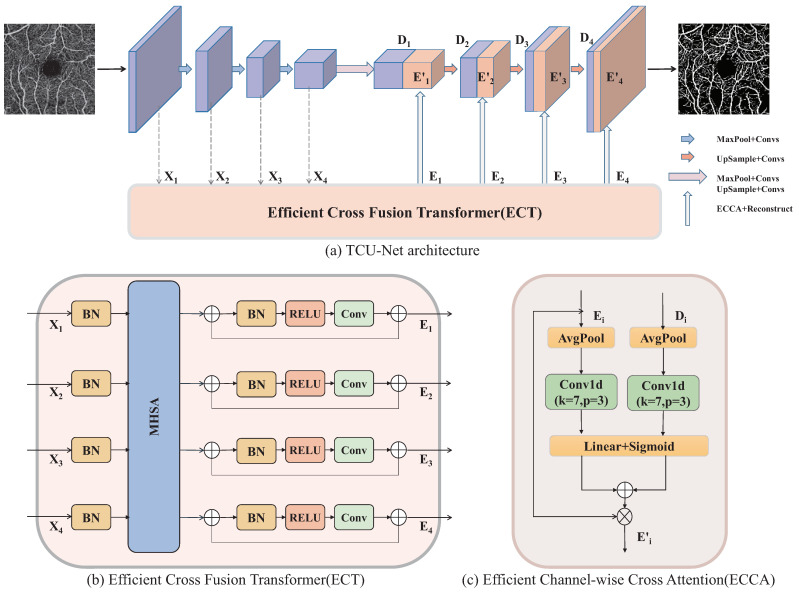
(**a**) Illustration of the proposed TCU-Net, (**b**) efficient cross-fusion transformer module, and (**c**) efficient channel cross-attention module.

**Figure 3 sensors-23-04897-f003:**
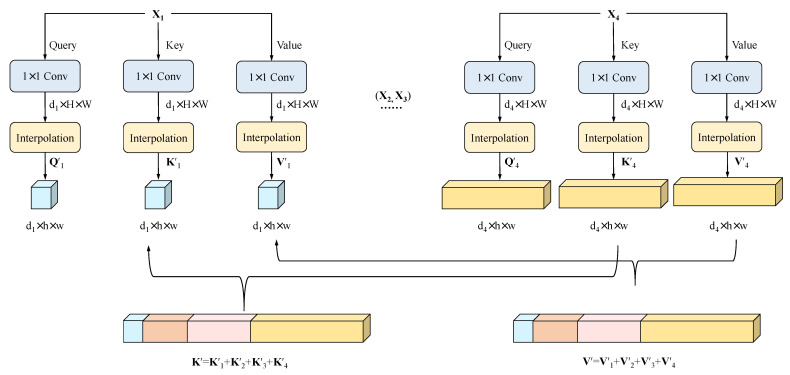
The encoder output is subjected to an interpolation downsampling operation to obtain the cross-scale ${Q^{\prime }}_{i}\left(i=1,2,3,4\right),K^{\prime },V^{\prime }$.

**Figure 4 sensors-23-04897-f004:**
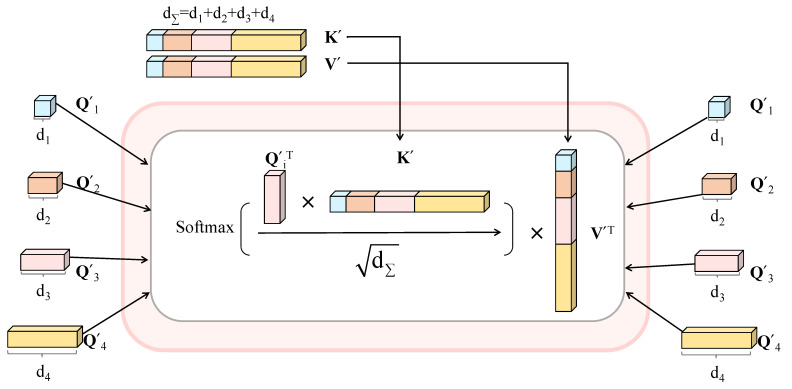
Efficient multihead cross-attention.

**Figure 5 sensors-23-04897-f005:**
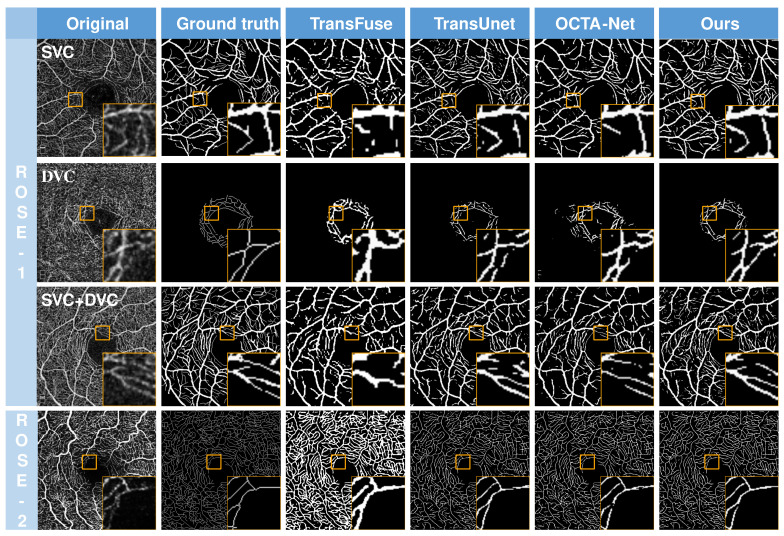
Vessel segmentation results from different methods on different layers of ROSE-1 and ROSE-2. From (**left**) to (**right**): en face angiograms (original images), manual annotations, and vessel segmentation results obtained by TransFuse, TransUnet, OCTA-Net, and the proposed method (TCU-Net), respectively.

**Figure 6 sensors-23-04897-f006:**
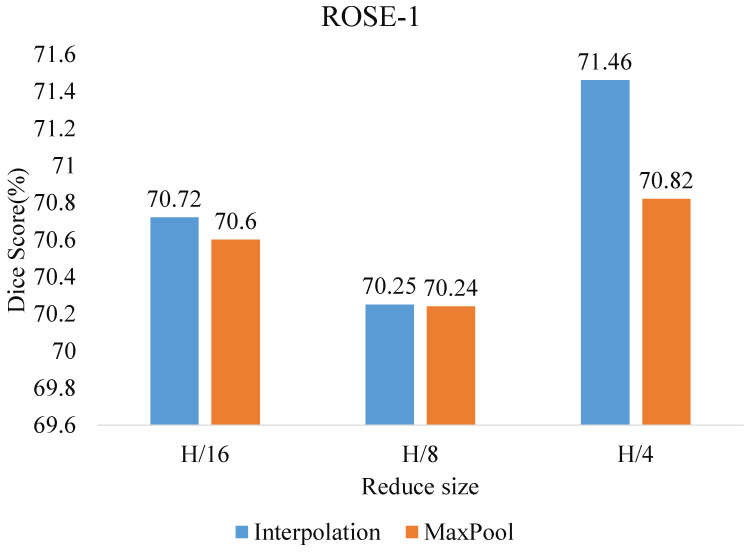
Effect of size reduction and projection of efficient self-attention on ROSE-1 dataset.

**Figure 7 sensors-23-04897-f007:**
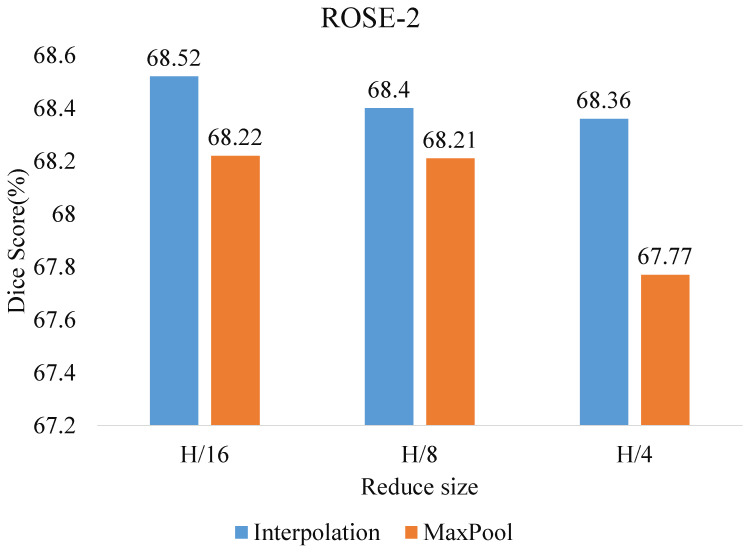
Effect of size reduction size and projection of efficient self-attention on ROSE-2 dataset.

**Table 1 sensors-23-04897-t001:** Quantitative results on ROSE-1 (**SVC**) datasets compared to previous SOTA. The evaluation metrics are based on the mean ± standard deviation calculated in repeated experiments. The values in bold denote the optimal results obtained under the present experimental conditions.

Methods	AUC (%)	ACC (%)	G-Mean (%)	Kappa (%)	Dice (%)	FDR (%)
U-Net [18]	94.10 ± 0.13	91.38 ± 0.18	82.52 ± 0.48	72.02 ± 0.20	77.48 ± 0.32	21.55 ± 1.62
ResU-Net [20]	94.57 ± 0.09	91.73 ± 0.12	84.26 ± 0.59	72.97 ± 0.32	77.05 ± 0.30	19.88 ± 1.58
CE-Net [40]	94.90 ± 0.07	91.63 ± 0.19	84.08 ± 0.49	71.71 ± 0.34	76.81 ± 0.24	19.57 ± 1.61
CS-Net [41]	95.07 ± 0.05	92.29 ± 0.07	83.41 ± 0.53	73.16 ± 0.22	77.78 ± 0.23	14.60 ± 1.14
OCTA-Net [7]	94.83 ± 0.12	92.09 ± 0.34	82.57 ± 1.54	72.24 ± 0.64	76.93 ± 0.59	14.17 ± 3.23
TranFuse [27]	92.50 ± 0.98	90.63 ± 0.50	83.09 ± 0.60	66.24 ± 0.28	72.64 ± 0.36	28.19 ± 3.20
TransUnet [26]	94.50 ± 0.11	92.21 ± 0.10	82.79 ± 0.48	72.18 ± 0.28	76.76 ± 0.26	12.34 ± 1.20
Ours	**95.12 ± 0.05**	**92.30 ± 0.06**	**84.73 ± 0.85**	**73.29 ± 0.31**	**77.91 ± 0.37**	**12.25 ± 2.11**

**Table 2 sensors-23-04897-t002:** Quantitative results on ROSE-1 (**DVC**) datasets compared to previous SOTA.

Methods	AUC (%)	ACC (%)	G-Mean (%)	Kappa (%)	Dice (%)	FDR (%)
U-Net [18]	95.33 ± 0.33	96.90 ± 1.23	84.87 ± 2.38	59.80 ± 3.70	60.70 ± 3.30	51.96 ± 4.20
ResU-Net [20]	96.65 ± 0.29	98.86 ± 0.12	89.76 ± 2.59	65.55 ± 4.32	61.12 ± 3.30	39.16 ± 4.58
CE-Net [40]	96.37 ± 0.39	98.08 ± 0.32	90.15 ± 2.72	64.30 ± 4.86	63.21 ± 3.18	51.81 ± 4.86
CS-Net [41]	96.65 ± 0.31	98.20 ± 1.09	89.30 ± 2.93	66.09 ± 4.26	66.18 ± 3.05	46.86 ± 4.17
OCTA-Net [7]	96.82 ± 0.56	98.29 ± 0.75	90.12 ± 2.63	64.07 ± 2.82	64.82 ± 2.64	46.71 ± 3.45
TranFuse [27]	94.95 ± 0.50	98.55 ± 1.27	85.89 ± 2.83	60.16 ± 3.88	60.86 ± 3.16	46.55 ± 3.14
TransUnet [26]	96.69 ± 0.05	99.01 ± 0.27	87.77 ± 2.60	68.96 ± 4.95	67.39 ± 3.26	33.21 ± 3.20
Ours	**98.23 ± 0.13**	**99.12 ± 0.20**	**90.23 ± 2.17**	**69.39 ± 2.96**	**69.87 ± 2.16**	**28.98 ± 3.11**

**Table 3 sensors-23-04897-t003:** Quantitative results on ROSE-1 (**SVC+DVC**) datasets compared to previous SOTA.

Methods	AUC (%)	ACC (%)	G-Mean (%)	Kappa (%)	Dice (%)	FDR (%)
U-Net [18]	90.17 ± 0.96	89.30 ± 0.30	77.58 ± 0.98	63.61 ± 0.12	69.77 ± 0.23	24.37 ± 1.71
ResU-Net [20]	91.14 ± 0.61	89.82 ± 0.38	77.84 ± 0.25	64.10 ± 0.25	70.12 ± 0.54	20.66 ± 3.52
CE-Net [40]	90.21 ± 0.04	89.63 ± 0.24	77.45 ± 0.91	63.44 ± 0.24	69.58 ± 0.27	21.08 ± 2.62
CS-Net [41]	91.49 ± 0.02	90.16 ± 0.06	77.47 ± 0.89	64.77 ± 0.44	70.52 ± 0.52	17.89 ± 1.59
OCTA-Net [7]	91.44 ± 0.05	90.12 ± 0.15	76.84 ± 0.99	64.31 ± 0.35	70.02 ± 0.47	17.14 ± 2.43
TranFuse [27]	89.86 ± 0.51	89.54 ± 0.26	76.51 ± 0.84	63.92 ± 0.78	68.96 ± 0.36	27.30 ± 3.51
TransUnet [26]	91.05 ± 0.05	90.15 ± 0.27	77.22 ± 0.60	64.56 ± 0.45	70.83 ± 0.67	16.93 ± 3.20
Ours	**91.70 ± 0.06**	**90.42 ± 0.16**	**77.98 ± 1.10**	**64.87 ± 0.38**	**71.20 ± 0.50**	**15.67 ± 2.56**

**Table 4 sensors-23-04897-t004:** Quantitative results on **ROSE-2** datasets compared to previous SOTA.

Methods	AUC (%)	ACC (%)	G-Mean (%)	Kappa (%)	Dice (%)	FDR (%)
U-Net [18]	85.03 ± 0.57	94.16 ± 0.19	79.39 ± 1.26	64.11 ± 0.50	67.33 ± 0.56	28.65 ± 1.90
ResU-Net [20]	86.08 ± 0.61	94.26 ± 0.88	80.12 ± 0.25	65.56 ± 0.25	68.75 ± 0.54	27.53 ± 1.52
CE-Net [40]	85.13 ± 0.06	94.03 ± 0.05	80.70 ± 0.29	65.71 ± 0.19	**69.04 ± 0.19**	27.76 ± 0.58
CS-Net [41]	85.98 ± 0.06	94.39 ± 0.20	78.20 ± 1.56	63.96 ± 0.61	67.02 ± 0.74	26.64 ± 2.02
OCTA-Net [7]	86.05 ± 0.04	94.44 ± 0.15	78.91 ± 0.74	64.92 ± 0.14	67.96 ± 0.16	**26.04 ± 0.14**
TranFuse [27]	84.01 ± 0.49	89.83 ± 0.26	79.94 ± 1.24	60.16 ± 0.68	66.03 ± 0.48	38.96 ± 2.84
TransUnet [26]	85.78 ± 0.05	94.24 ± 0.27	79.91 ± 0.60	63.97 ± 0.95	68.14 ± 0.26	27.77 ± 1.20
Ours	**86.23 ± 0.05**	**94.54 ± 0.24**	**81.26 ± 0.62**	**64.97 ± 0.21**	68.40 ± 0.28	**25.26 ± 1.24**

**Table 5 sensors-23-04897-t005:** Comparison of parameters and floating point of operations in above methods.

Methods	Param (M)	FLOPs (G)
U-Net [18]	34.5	184.6
ResU-Net [20]	12.0	22.1
CE-Net [40]	29.0	25.7
CS-Net [41]	33.6	157.2
OCTA-Net [7]	217.7	345.0
TranFuse [27]	300.16	420.6
TransUnet [26]	334.18	483.4
Ours	14.1	80.6

**Table 6 sensors-23-04897-t006:** Ablation studies on ROSE-1 (**SVC**) dataset.

Methods	AUC (%)	ACC (%)	G-Mean (%)	Kappa (%)	Dice (%)	FDR (%)
Baseline (U-Net)	94.10	91.38	82.52	72.02	77.48	21.55
Baseline+ECT	95.10	92.38	84.21	73.77	78.35	15.37
Baseline+ECCA	95.01	92.36	83.23	73.25	77.79	13.91
Baseline+ECT+ECCA	**95.11**	**92.39**	**84.73**	**73.78**	**78.45**	**12.25**

**Table 7 sensors-23-04897-t007:** Ablation studies on ROSE-1 (**DVC**) dataset.

Methods	AUC (%)	ACC (%)	G-Mean (%)	Kappa (%)	Dice (%)	FDR (%)
Baseline (U-Net)	95.33	96.90	84.87	59.80	60.70	51.96
Baseline+ECT	98.29	99.02	**91.84**	64.83	65.44	43.61
Baseline+ECCA	98.00	98.63	90.37	69.93	70.41	36.67
Baseline+ECT+ECCA	**98.40**	**99.18**	90.03	**71.40**	**71.80**	**26.51**

**Table 8 sensors-23-04897-t008:** Ablation studies on ROSE-1 (**SVC+DVC**) dataset.

Methods	AUC (%)	ACC (%)	G-Mean (%)	Kappa (%)	Dice (%)	FDR (%)
Baseline (U-Net)	90.17	89.30	77.58	63.61	69.77	24.37
Baseline+ECT	91.39	90.16	78.15	65.17	70.97	18.93
Baseline+ECCA	91.60	90.19	77.61	64.91	70.64	17.97
Baseline+ECT+ECCA	**91.76**	**90.31**	**79.21**	**65.52**	**71.46**	**17.34**

**Table 9 sensors-23-04897-t009:** Ablation studies on **ROSE-2** dataset.

Methods	AUC (%)	ACC (%)	G-Mean (%)	Kappa (%)	Dice (%)	FDR (%)
Baseline (U-Net)	85.03	94.16	79.39	64.11	67.33	28.65
Baseline+ECT	86.20	94.38	79.09	64.72	67.78	26.65
Baseline+ECCA	86.19	94.21	78.93	64.93	67.99	28.29
Baseline+ECT+ECCA	**86.29**	**94.43**	**80.10**	**64.97**	**68.15**	**26.51**

## Data Availability

Publicly available datasets were used in this study. The Retinal OCTA Segmentation dataset can be found here: https://imed.nimte.ac.cn/dataofrose.html (accessed on 23 March 2023).

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
