# Peer review of "TCU-Net: Transformer Embedded in Convolutional U-Shaped Network for Retinal Vessel Segmentation"

_sensors, 2023, doi:10.3390/s23104897_

Round 1
Reviewer 1 Report
In this paper, authors propose a self-attentive mechanism with linear complexity, called TCU network, which combines transformers and 2D convolutional neural networks to segment retinal vessels. Authors did an interesting work and the following review comments are recommended, and the authors are invited to explain and modify them.
1 Abstract is unnecessarily wordy. Make it brief and concise. In addition, Conclusion should clearly state the outcome. Some of the obtained results need to be highlighted in the conclusion section.
2 “In this paper, we introduce the Transformer Embedded in Convolutional U-shaped Network”, novelty is confusing. The main contributions of the manuscript are not clear. The main contributions of the article must be very clear and would be better if summarize them into 3-4 points.
3 The introduction section is very weak. An introduction is an important road map for the rest of the paper and should consist of an opening hook to catch the researcher's attention, relevant background studies, and a concrete statement that presents the main argument, but your introduction lacks these fundamentals, especially relevant background studies. This related work is just listed out without comparing the relationship between this paper's model and theirs; only the method flow is introduced at the end, and the principle of the method is not explained. To make soundness of your study must include these latest related works and discuss them.
FIVES: A Fundus Image Dataset for Artificial Intelligence based Vessel Segmentation. Scientific Data, 10.1038/s41597-022-01564-3
Automatic interpretation and clinical evaluation for fundus fluorescein angiography images of diabetic retinopathy patients by deep learning. British Journal of Ophthalmology, 10.1136/bjo-2022-321472
A clinical decision model based on machine learning for ptosis. BMC Ophthalmology, 10.1186/s12886-021-01923-5
4 “Initial learning rate of 0.0006 and weight decay of 0.0001. The batch size is set as 2”, how to optimize these hyperparameters during model training?
5 Authors did not mention implementation challenges.
6 When writing phrases like “The evaluation metrics are calculated as follows:” it must cite related work in order to sustain the statement (https://doi.org/10.1155/2023/2345835).
7 Could you please check your references carefully? All references must be complete before the acceptance of a manuscript.
8 Moreover, it should be noticed that the clinical appliance has to be decided by medicals since the existing differences between the real image and the one generated by the proposed model could be substantial in the medical field.
Author Response
1. The reviewer’s comment: Abstract is unnecessarily wordy. Make it brief and concise. In addition, Conclusion should clearly state the outcome. Some of the obtained results need to be highlighted in the conclusion section.
The authors’ Answer: Thanks to your suggestions, we have rewritten the Abstract section to remove unnecessary explanations according to the reviewers' suggestions. In addition, the experimental results obtained by the proposed method are explained in the Conclusion section.
2. The reviewer’s comment:“In this paper, we introduce the Transformer Embedded in Convolutional U-shaped Network”, novelty is confusing. The main contributions of the manuscript are not clear. The main contributions of the article must be very clear and would be better if summarize them into 3-4 points.
The authors’ Answer: In the second paragraph of Introduction, line 67, we analyze the advantages and disadvantages of existing schemes for color fundus images and OCTA images, and add a summary of the shortcomings of existing methods for OCTA image segmentation in line 123. To address these shortcomings, we introduce our method TCU-Net, and summarize the innovation points at the end of the Introduction.
3. The reviewer’s comment:The introduction section is very weak. An introduction is an important road map for the rest of the paper and should consist of an opening hook to catch the researcher's attention, relevant background studies, and a concrete statement that presents the main argument, but your introduction lacks these fundamentals, especially relevant background studies. This related work is just listed out without comparing the relationship between this paper's model and theirs; only the method flow is introduced at the end, and the principle of the method is not explained. To make soundness of your study must include these latest related works and discuss them.
The authors’ Answer:
- We have rewritten the Introduction section based on your advice. In the first sentence of the first paragraph of line 40, the importance of retinal vascular images for ophthalmic diseases is introduced, and the fundus vascular imaging features of the OCTA technique are introduced based on this status.
- In the second paragraph of Introduction section, we introduce the existing color fundus image segmentation method, and then add the principle of skip connection in U-Net network, and analyze the advantages and disadvantages of the improvement of skip connection by Attention U-Net, which provides an interpretable analysis for our innovation "ECT module" to replace the skip connection of U-Net.
- In the third paragraph of Introduction section, we add the latest existing methods for OCTA fundus image segmentation. We also compare the advantages and disadvantages of the existing methods and provide an explanation for our use of Transformer model. Based on the suggestions of the latest related works given by the reviewers, we introduce and discuss the related works for OCTA retinal vascular segmentation. The citations added in the Introuction section and Related Work section are as follows:
Pissas, Theodoros, et al. "Deep iterative vessel segmentation in OCT angiography." Biomedical Optics Express 11.5 (2020): 2490-2510.
Li, Mingchao, Weiwei Zhang, and Qiang Chen. "Image magnification network for vessel segmentation in octa images." Pattern Recognition and Computer Vision: 5th Chinese Conference, PRCV 2022, Shenzhen, China, November 4–7, 2022, 2022, Proceedings, Part IV. Cham: Springer Nature Switzerland, 2022.
Tan, Xiao, et al. "OCT2Former: A retinal OCT-angiography vessel segmentation transformer." Computer Methods and Programs in Biomedicine 233 (2023): 107454.
Xu, Xiayu, et al. "AV-casNet: Fully Automatic Arteriole-Venule Segmentation and Differentiation in OCT Angiography." IEEE Transactions on Medical Imaging (2022).
Wu, Zhuojie, et al. "PAENet: A progressive attention-enhanced network for 3D to 2D retinal vessel segmentation." 2021 IEEE International Conference on Bioinformatics and Biomedicine (BIBM). IEEE, 2021.
4. The reviewer’s comment: “Initial learning rate of 0.0006 and weight decay of 0.0001. The batch size is set as 2”, how to optimize these hyperparameters during model training?
The authors’Answer: We use the Adam optimizer to optimize the hyperparameters, as modified in Section 4.2 Implements Details.
5. The reviewer’s comment: Authors did not mention implementation challenges.
The authors’ Answer: Since we trained on the existing OCTA dataset and designed a neural network with linear computational complexity, the inference of the model is faster and the memory footprint is smaller compared to a traditional Transformer network. Therefore, it can be carried out simply in the experimental phase.
6. The reviewer’s comment: When writing phrases like “The evaluation metrics are calculated as follows:” it must cite related work in order to sustain the statement (https://doi.org/10.1155/2023/2345835).
The authors’ Answer: We apologize for not taking into account that the evaluation metrics cite related work as support, and in ‘4.1 Datasets and Metrics’ section we have added the following related work:
Bradley, Andrew P. "The use of the area under the ROC curve in the evaluation of machine learning algorithms." Pattern recognition 30.7 (1997): 1145-1159.
Diebold, Francis X., and Robert S. Mariano. "Comparing predictive accuracy." Journal of Business & economic statistics 20.1 (2002): 134-144.
McHugh, Mary L. "Interrater reliability: the kappa statistic." Biochemia medica 22.3 (2012): 276-282.
Benjamini, Yoav, and Yosef Hochberg. "Controlling the false discovery rate: a practical and powerful approach to multiple testing." Journal of the Royal statistical society: series B (Methodological) 57.1 (1995): 289-300.
Shamir, Reuben R., et al. "Continuous dice coefficient: a method for evaluating probabilistic segmentations." arXiv preprint arXiv:1906.11031 (2019).
7. The reviewer’s comment : Could you please check your references carefully? All references must be complete before the acceptance of a manuscript.
The authors’ Answer: Thank you for pointing this out, we have gone through the references and inserted the newly added references.
8.The reviewer’s comment: Moreover, it should be noticed that the clinical appliance has to be decided by medicals since the existing differences between the real image and the one generated by the proposed model could be substantial in the medical field.
The authors’ Answer: Thanks to your comments, we have added the limitations of the proposed model in the actual medical field in the Conclusion section.
Reviewer 2 Report
Dear Authors,
The manuscript entitled with "TCU-Net: Transformer Embedded in Convolutional U-shaped Network for Retinal Vessel Segmentation" proposed a TCU-Net method for retinal vessel segmentation. In their method, they proposed an Efficient Cross-scale Transformer (ECT) module and an Efficient Channel Cross-Attention (ECCA) module. They validate the effectiveness of TCU-Net method on ROSE dataset. They also validate the proposed ECT and ECCA modules through ablation exepriment. The idea is interesting. The work is good. However, I have a suggestion for the authors. The authors should outstand their contributions of their work in the Introduction section.
Author Response
The reviewer’s comment:The manuscript entitled with "TCU-Net: Transformer Embedded in Convolutional U-shaped Network for Retinal Vessel Segmentation" proposed a TCU-Net method for retinal vessel segmentation. In their method, they proposed an Efficient Cross-scale Transformer (ECT) module and an Efficient Channel Cross-Attention (ECCA) module. They validate the effectiveness of TCU-Net method on ROSE dataset. They also validate the proposed ECT and ECCA modules through ablation exepriment. The idea is interesting. The work is good. However, I have a suggestion for the authors. The authors should outstand their contributions of their work in the Introduction section.
The authors’ Answer: Thanks to your professional advice, we have added four contributions about our work at the end of the Introduction for the reader to read.
Reviewer 3 Report
Dear Editor and Dear Authors,
The topic of this paper is very interesting and thank you for your trust to send me the manuscript to review it.
The manuscript proposes a novel self-attentive mechanism with linear complexity, called TCU network, which combines transformers and 2D convolutional neural networks to segment retinal vessels. Authors have evaluated TCU-Net on the dedicated Retinal OCTA Segmentation (ROSE) dataset, and they claim that the experiments show that their proposed method produces superior vessel segmentation performance and robustness against the state-of-the-art approaches.
The comments and suggestions are below:
1. The use of personal pronounses (we, our, etc.) is unacceptable for scientific English. Please use Passive Voice instead in the whole paper. This mandatory.
2. The Abstract contains abbreviations, it is repulsive for readers. This is unusual, since the abbreviations should be explained later in the text at first appearance. Also, not all abbreviations are explained in the abstract itself.
3. The content of the Abstract is not poor, however it should be rewritten in Passive Voice (and the whole paper too) with more concise wording and maybe some numerical results should be given in the end.
4. The Introduction should be improved. Authors should point out their contribution with more concise wording. The advantages and drawbacks of the method should be mentioned in brief. Finally in the end of the section, a brief summary of the paper itself should be given.
5. The related works section should be improved. It is not enough to mention the used methods only in brief in corresponding references, authors should mention the results, advantages, or drawbacks of cited methods. Maybe in the end of the section a qualitative comparison of the cited references could be given.
6. The proposed TCU-Net in Figure 2 should be explained with more details. Since it relies on U-Net, the U-Net should be explained too with sufficient details, and the differences with regard to the proposed method.
7. The results section is satisfactory. It contains sufficient details and explanations.
8. I think the Conclusion section should be extended with more explanations (i.e. drawback vs advantages).
9. The English Language and the grammar should be checked.
The paper is very interesting from the practical point of view, however it has drawbacks that should be corrected properly.
Recommendation: revision.
Author Response
1.The reviewer’s comment: The use of personal pronounses (we, our, etc.) is unacceptable for scientific English. Please use Passive Voice instead in the whole paper. This mandatory.
The authors’ Answer: We have done our best to improve the manuscript and have made some changes to it. These changes do not affect the content or the framework of the paper. We do not list the changes here, but use blue markers in the revised paper.
2. The reviewer’s comment: The Abstract contains abbreviations, it is repulsive for readers. This is unusual, since the abbreviations should be explained later in the text at first appearance. Also, not all abbreviations are explained in the abstract itself.
The authors’ Answer: We apologize for the high number of abbreviations in the abstract, and we have removed all of them in the reworked and revised abstract.
3. The reviewer’s comment: The content of the Abstract is not poor, however it should be rewritten in Passive Voice (and the whole paper too) with more concise wording and maybe some numerical results should be given in the end.
The authors’ Answer: Thanks for pointing out that we have removed the detailed explanation for convolutional networks from the abstract (a specific explanation exists in the text) and introduced our innovative module using the passive voice. Finally, we have added numerical results at the end.
4. The reviewer’s comment: The Introduction should be improved. Authors should point out their contribution with more concise wording. The advantages and drawbacks of the method should be mentioned in brief. Finally in the end of the section, a brief summary of the paper itself should be given.
The authors’ Answer: Based on your suggestions, we have rewritten the Introduction section, starting with the second paragraph of Introduction, line 67, which describes the strengths and weaknesses of existing work related to fundus vascular segmentation and summarizes the problems that need improvement in the fifth paragraph, line 123. The improvements proposed by our method to address these issues are presented in the last paragraph, and a summary of the contributions made to the paper is provided.
5. The reviewer’s comment: The related works section should be improved. It is not enough to mention the used methods only in brief in corresponding references, authors should mention the results, advantages, or drawbacks of cited methods. Maybe in the end of the section a qualitative comparison of the cited references could be given.
The authors’ Answer: In Related work, we insert an analysis of existing retinal vessel segmentation schemes, discuss the advantages and disadvantages of these works, and give a qualitative analysis at the end.
6. The reviewer’s comment: The proposed TCU-Net in Figure 2 should be explained with more details. Since it relies on U-Net, the U-Net should be explained too with sufficient details, and the differences with regard to the proposed method.
The authors’ Answer: In section ‘3.1 Network Architecture’, we added an introduction to the framework structure of U-Net and explained the differences between our framework and U-Net.
7. The reviewer’s comment: The results section is satisfactory. It contains sufficient details and explanations.
The authors’ Answer: Thanks for the reviewer’s positive comments.
8. The reviewer’s comment: I think the Conclusion section should be extended with more explanations (i.e. drawback vs advantages).
The authors’ Answer: We have enriched the closing section with your suggestions to illustrate the limitations of our program.
9. The reviewer’s comment:The English Language and the grammar should be checked.
The authors’ Answer: We have optimized our language and syntax as much as possible for the reader to access.
Round 2
Reviewer 1 Report
Authors tried to revise the manuscript, but there are still some concerns; authors should carefully read comments and revise the paper accordingly.
1 “Initial learning rate of 0.0006 and weight decay of 0.0001. The batch size is set as 2”, how to optimize these hyperparameters during model training?
Authors ‘Answer: “We use the Adam optimizer to optimize the hyperparameters”.
Concern:
Hyperparameters are parameters whose values are set before starting the model training process such as hidden layers, number of epochs, learning rate, momentum, batch size etc. what strategy/search was used to optimize these hyperparameters?
2 The introduction section is very weak. An introduction is an important road map for the rest of the paper and should consist of an opening hook to catch the researcher's attention, relevant background studies, and a concrete statement that presents the main argument, but your introduction lacks these fundamentals, especially relevant background studies. This related work is just listed out without comparing the relationship between this paper's model and theirs; only the method flow is introduced at the end, and the principle of the method is not explained. To make soundness of your study must include these latest related works and discuss them.
FIVES: A Fundus Image Dataset for Artificial Intelligence based Vessel Segmentation. Scientific Data, 10.1038/s41597-022-01564-3
Automatic interpretation and clinical evaluation for fundus fluorescein angiography images of diabetic retinopathy patients by deep learning. British Journal of Ophthalmology, 10.1136/bjo-2022-321472
A clinical decision model based on machine learning for ptosis. BMC Ophthalmology, 10.1186/s12886-021-01923-5
The authors’ Answer: “We have rewritten the Introduction section based on your advice.”
Concern:
Where is the comparing the relationship and where is the suggested studies in revised version?
3 When writing phrases like “The evaluation metrics are calculated as follows:” it must cite related work in order to sustain the statement (https://doi.org/10.1155/2023/2345835)
Concern:
The evaluation metrics used for medical image segmentation are the same as in the suggested reference, so it was better to just cite the suggested work to sustain the statement.
Authors cite the wrong reference, for example, "Dice coefficient (Dice) [39]";
[39] Shamir, R.R.; Duchin, Y.; Kim, J.; Sapiro, G.; Harel, N. Continuous dice coefficient: a method for evaluating probabilistic segmentations. arXiv preprint arXiv:1906.11031 2019.
The correct reference for DICE: “L. R. Dice, ‘‘Measures of the amount of ecologic association between species,’’ Ecology, vol. 26, no. 3, pp. 297–302, 1945”.
4 Moreover, it should be noticed that the clinical appliance has to be decided by medicals since the existing differences between the real image and the one generated by the proposed model could be substantial in the medical field.
The authors’ Answer: Thanks to your comments, we have added the limitations of the proposed model in the actual medical field in the Conclusion section.
Concern: Authors should carefully revise.
Author Response
We feel great thanks for your professional review work on our article. As you are concerned, there are several problems that need to be addressed. According to your nice suggestions, we have made extensive corrections to our previous draft. Our response is given in normal font and changes/additions to the manuscript are given in the red text.
1. The reviewer’s comment:Hyperparameters are parameters whose values are set before starting the model training process such as hidden layers, number of epochs, learning rate, momentum, batch size etc. what strategy/search was used to optimize these hyperparameters?
The authors’ Answer: We apologize for the previous error. In '4.2 Implements Details' section, we have added a hyperparameter optimization method that uses a random search strategy to find the optimal hyperparameter.
2.The reviewer’s comment:Where is the comparing the relationship and where is the suggested studies in revised version?
The authors’ Answer: In line 68 of the second paragraph of the 'Introduction' section, we marked the citations recommended by the reviewers in red font and added the analysis of these methods in the application of medical segmentation.
Gao, Z.; Pan, X.; Shao, J.; Jiang, X.; Su, Z.; Jin, K.; Ye, J. Automatic interpretation and clinical evaluation for fundus fluorescein 569 angiography images of diabetic retinopathy patients by deep learning. British Journal of Ophthalmology 2022.
Jin, K.; Huang, X.; Zhou, J.; Li, Y.; Yan, Y.; Sun, Y.; Zhang, Q.; Wang, Y.; Ye, J. Fives: A fundus image dataset for artificial 571 Intelligence based vessel segmentation. Scientific Data 2022, 9, 475.
Song, X.; Tong, W.; Lei, C.; Huang, J.; Fan, X.; Zhai, G.; Zhou, H. A clinical decision model based on machine learning for ptosis. 573 BMC ophthalmology 2021, 21, 1–9.
3. The reviewer’s comment:The evaluation metrics used for medical image segmentation are the same as in the suggested reference, so it was better to just cite the suggested work to sustain the statement. Authors cite the wrong reference, for example, "Dice coefficient (Dice) [39]"; [39] Shamir, R.R.; Duchin, Y.; Kim, J.; Sapiro, G.; Harel, N. Continuous dice coefficient: a method for evaluating probabilistic segmentations. arXiv preprint arXiv:1906.11031 2019.
The correct reference for DICE: “L. R. Dice, ‘‘Measures of the amount of ecologic association between species,’’ Ecology, vol. 26, no. 3, pp. 297–302, 1945”.
The authors’ Answer: Thanks for your suggestion, we replaced the quotation about Dice coefficient (Dice) in line 402 of the section '4.1 Datasets and Metrica'. The citation is updated to :
Dice, L.R. Measures of the amount of ecologic association between species. Ecology 1945, 26, 297–302
4. The reviewer’s comment: Moreover, it should be noticed that the clinical appliance has to be decided by medicals since the existing differences between the real image and the one generated by the proposed model could be substantial in the medical field. Authors should carefully revise.
The authors’ Answer: Thank you very much for pointing out that we have carefully revised the conclusion and optimized the logic.
Reviewer 3 Report
The paper is significantly improved, however the paper is not corrected completely to Passive Voice.
Recommendation: accept
Author Response
Dear reviewers,
We feel great thanks for your professional review work on our article. As you are concerned, there are several problems that need to be addressed. According to your nice suggestions, we have made extensive corrections to our previous draft. Our response is given in normal font and changes/additions to the manuscript are given in the red text.
1. The reviewer’s comment: The paper is significantly improved, however, the paper is not corrected completely to Passive Voice.
The authors’ Answer: Thank you for your feedback. We appreciate your help in making our paper more clear and more professional and have optimized the language in the paragraphs marked in red and reduced the use of Active Voice.